# Impact of Provincial Income Inequality on Parenting Styles in China during COVID-19

**DOI:** 10.3390/bs14070587

**Published:** 2024-07-10

**Authors:** Rui Jin, Na Liu, Hao Zhou, Mingren Zhao

**Affiliations:** 1Faculty of Education, Shenzhen University, Shenzhen 518060, China; rji227@szu.edu.cn (R.J.); haozhou@szu.edu.cn (H.Z.); 2Faculty of College of Foreign Languages, Zhejiang Ocean University, Zhoushan 316022, China; na.liu@zjou.edu.cn

**Keywords:** parenting styles, income inequality, CFPS, latent class analysis

## Abstract

Research on Chinese parenting styles using representative samples is limited, particularly during COVID-19, with most studies focusing on individual factors while neglecting regional influences. This study examines the impact of provincial income inequality, measured by the Gini coefficient, on parenting styles and how these effects vary across subgroups. Using data from the China Family Panel Studies (CFPS) 2020, encompassing 3768 children aged 7–16 years from 25 regions, we employed a multinomial logistic regression model to analyze the predictability of provincial income inequality on parenting styles identified through latent class analysis. Three parenting styles emerged during the first year of COVID-19: authoritarian (48.2%), autonomy granting (27.7%), and average-level undifferentiated (24.1%). A higher Gini coefficient related to a greater likelihood of parents adopting authoritarian or autonomy-granting parenting styles over average-level undifferentiated parenting. Subgroup analyses revealed a higher likelihood of adopting autonomy-granting and authoritarian parenting for male children compared to female. Mothers with lower education levels and parents in rural areas tend to favor authoritarian parenting in response to higher income inequality. This trend was less evident among more educated mothers and parents living in urban areas. These findings suggest that parenting styles in China are influenced by complex and region-specific factors.

## 1. Introduction

The COVID-19 outbreak drastically altered family education in China in 2020, with strict isolation leading to significant changes. On the one side, families, confined to small spaces, faced increased restlessness among children and mounting social pressures as parents balanced telework with full-time childcare, all without external support. These conditions have led to several family issues, such as exacerbated generational communication gaps [1], increased parent burnout [2], heightened childcare demands [3], and an increase in depression and anxiety [4].

On the other hand, it has unexpectedly provided an opportunity for positive transformations in family dynamics. With the enforced increase in time spent at home due to quarantine measures, families have had the chance to reevaluate and prioritize their family life, potentially strengthening bonds and fostering a greater sense of cohesion [5]. These fluctuations, both detrimental and beneficial, have influenced how parents guide and interact with their children, pointing to the crucial role of parenting style in nurturing children’s skill development during a global health crisis [6]. Therefore, it is imperative to explore the spectrum of parenting styles during this challenging period in China; however, research in this area, particularly in the Chinese context, has been limited.

In addition, the prior literature has shown that a variety of resources at parents’ disposal, such as social networks [7], cultural assets [8], and political connections [9], significantly influence parenting style choices. However, except for these individual-level factors, Doepke and Zilibotti [10] innovatively put forward that economic conditions, specifically income inequality, emerge as a crucial determinant of parenting methods. In societies marked by lower income inequality, there is a tendency towards more permissive parenting styles. These arguments and findings prompted us to explore the dynamics within the Chinese context. As the largest developing country, China has witnessed unparalleled economic growth but with large economic developmental disparities across provinces [11]. Therefore, it is interesting to examine how provincial income inequality in China influences parenting styles.

Finally, the impact of income inequality on students may differ significantly across various subgroups. In particular, the cultural preference for sons over daughters significantly influences educational investments and parenting styles, a dynamic that is far less prominent in Western societies [12,13]. Also, the traditional urban–rural divide, which shapes access to educational resources and opportunities, represents a unique aspect of the Chinese socioeconomic structure not commonly observed to the same extent in other countries [14,15]. Moreover, within these varied contexts, mothers in China often assume a disproportionately larger share of parenting responsibilities, a gendered distribution of domestic labor that has significant implications for the educational guidance children receive [16]. Therefore, exploring these subgroup differences in the context of provincial income inequality provides valuable insights into the complex interplay between economic factors and family dynamics, offering crucial implications for educational policy and intervention strategies.

Building upon these foundational insights and unresolved questions, this study shifts the focus from the well-explored individual-level factors to the broader context of provincial income inequality in China during the first year of the COVID-19 pandemic. It aims to investigate the variety of parenting styles that emerged during this challenging period and to examine whether Doepke and Zilibotti’s [10] assertions regarding the impact of income inequality on parenting are applicable to the Chinese context. Utilizing data from the China Family Panel Studies (CFPS) conducted between July and December 2020, this study analyzes responses from 3768 children spanning 25 provinces and regions. We first identified latent classes of parenting styles. Then, we examined the predictability of a provincial-level factor (i.e., provincial income inequality) on the identified latent classes, followed by disaggregating the results by subgroups: child gender, maternal education level, and hukou status (household registration; rural vs. urban). The research questions are as follows:How many latent styles of Chinese parenting can be identified during the first year of the COVID-19 pandemic?What influences does provincial income inequality have on the probability of being in an identified parenting style?What are the differences in the effects of provincial income inequality across the various subgroups (i.e., child gender, maternal education level, and hukou status)?

## 2. Literature Review

### 2.1. Patterns of Parenting Styles

Baumrind [17,18] initially outlined three parenting styles: authoritative, authoritarian, and permissive. McCoby and Martin [19] later refined this framework, introducing a two-facet model emphasizing “demandingness” and “responsiveness” and adding a fourth style: neglectful. Authoritative parenting combines high expectations with responsiveness, fostering assertiveness and cooperation in children. Authoritarian parents are demanding but unresponsive, emphasizing obedience through strict control. Permissive parenting is characterized by high responsiveness but low demands, with minimal control exerted by parents. Neglectful parenting lacks both demandingness and responsiveness, resulting in disengagement and potential neglect.

Research with Western participants has validated authoritative, authoritarian, and neglectful parenting styles, but permissive parenting lacks empirical support [9]. Compared to Western research, empirical studies on parenting styles in China are limited. For example, authoritative and authoritarian styles are predominant among Chinese parents [9]. Zhang et al. [20] supported these findings, highlighting that these styles align with traditional Chinese cultural values that emphasize respect for authority, discipline, and academic achievement. Chao [21] further noted that while authoritative parenting is effective in Western contexts, Chinese parenting often incorporates high control and high warmth, suggesting a blend of care/autonomy with traditional authoritative and authoritarian elements. Permissive or neglectful parenting, though included in Baumrind’s framework, appears to be rare in the Chinese context, possibly due to the strong cultural emphasis on parental involvement in a child’s education and success [6,9].

Moreover, studies conducted in mainland China suggest that while Baumrind’s parenting styles provide a useful starting point, they might not fully capture the nuances of Chinese parenting. For instance, while studies identify parenting styles consistent with Baumrind’s authoritative and authoritarian categories, they also reveal a distinct parenting style in China termed “strict–affectionate” [20]. This style, characterized by high levels of warmth, encouragement of independence, supervision, and even harshness, shares features with Baumrind’s categories but represents a unique combination not fully encompassed by her original typology. It reflects the Chinese concept of Yanci, which emphasizes a balance of strictness and kindness in parenting. This dual approach combines elements of both authoritativeness and authoritarianism [20], highlighting the importance of discipline and emotional support. Similar to the tiger parenting identified by Kim et al. [22], strict–affectionate parenting also scores high on warmth and encouragement of independence. However, there are notable differences: Strict–affectionate mothers demonstrate much higher levels of monitoring, whereas tiger mothers exhibit higher levels of hostility. Furthermore, strict–affectionate parenting is associated with positive academic performance, emotional well-being, and family relationships, which contrasts with the generally negative outcomes linked to tiger parenting in adolescents.

More recently, a review has discovered that parenting styles need to incorporate dimensions of care and autonomy, revealing complex interactions between parental warmth, autonomy support, and psychological control [23]. These styles highlight the importance of autonomy-supportive environments for promoting adaptive psychosocial functioning and academic success among adolescents. These studies suggest that traditional parenting styles may not fully capture the diversity of parenting practices observed in contemporary Chinese families. Additionally, Huang’s review [23] and previous studies, e.g., [9,20,24], show that most Chinese studies rely on regional datasets, with few utilizing nationally representative samples, especially during the COVID-19 pandemic. These limitations underscore the need for more extensive research using nationwide samples to better understand the full spectrum of parenting practices in China.

### 2.2. Chinese Parenting during COVID-19

Existing studies indicate that parenting responses to COVID-19 exhibit mixed outcomes. On the negative side, research indicates that during the pandemic, over half of the caregivers reported engaging in potentially negative behaviors such as neglect or harsh parenting [25]. In The Netherlands, a study found a noticeable increase in severe parenting practices, such as more frequent shouting and derogatory remarks during the lockdown compared to pre-pandemic times [26]. Similarly, increased instances of parent–child conflicts, yelling at children, and physical punishments were reported [27,28].

On the positive side, the COVID-19 pandemic unexpectedly enhanced family dynamics, as parents spent more time at home due to quarantine measures [5,29]. This increased presence allowed parents to engage more deeply with their children through various activities that might not have been feasible previously. For instance, parents became more involved in their children’s educational activities, assisting with schoolwork and facilitating home-based learning. Recreational activities also saw a significant boost, with families spending more time playing games, engaging in creative projects, and exploring hobbies together [30,31]. Particularly for children with special needs, parents discovered new strengths in their children, which deepened their understanding and improved their parenting approaches [32]. This period also prompted parents to reevaluate and prioritize family life, empowering them to support their children’s future development [33]. Many parents began to appreciate the often-overlooked privileges of parenting, leading to a renewed sense of fulfillment and connection within the family unit [27].

Applying these to the Chinese context, we may expect variations from the traditional four parenting styles (i.e., authoritarian, authoritative, permissive, and neglectful). In the first year of COVID-19, many nations opted for a less stringent approach, closing only non-essential enterprises like bars and restaurants, while keeping schools and factories operational to mitigate early economic repercussions from the pandemic [34,35,36]. However, Chinese authorities implemented stricter quarantine measures in affected cities, mandating residents to isolate at home, suspending educational and occupational activities, and restricting travel [37,38]. Residents even were required to present a daily updated nucleic acid code for outdoor activities, which imposed significant barriers to interpersonal interactions [39].

Consequently, these substantial shifts may have influenced the practices and psychological approaches of Chinese parents. The stringent quarantine measures increased the time parents and children spent together, which, combined with economic pressure such as job losses and reduced incomes, heightened parental stress and anxiety [1,2]. In response, many parents may have increased demands on their children, leading to a predominance of authoritarian practices. Research has shown that heightened stress levels can exacerbate controlling behaviors in parents as they seek to maintain order and predictability in uncertain times [5]. However, the enforced home confinement also meant that parents spent more time with their children [32], likely reducing the prevalence of neglectful parenting. This aligns with findings from other countries (e.g., the United States, Japan, and Korea) where increased parental presence led to more engaged and attentive parenting behaviors [30,31], despite the stressful circumstances. Accordingly, we proposed the following hypotheses.

**H1.** *Authoritarian parenting will be a prominent style among Chinese parents during the COVID-19 pandemic*.

**H2.** *The prevalence of neglectful parenting will be low during the COVID-19 pandemic*.

Moreover, our study introduces a third dimension—autonomy-granting—to assess parenting [40,41], prompted by two key considerations. First, this trend towards fostering autonomy aligns with broader socio-economic shifts observed in recent research. The transition from a state-socialist to a market-oriented economy has been found to encourage a shift from traditional to modern parenting values in contemporary China [39]. In environments where creativity and initiative are rewarded, traditional attributes such as obedience and humbleness are being re-evaluated [42]. Consequently, this has led Chinese parents to increasingly adopt an autonomy-granting parenting style that emphasizes the individuality and autonomy of their children [43]. Second, while Doepke and Zilibotti [10] observed a general negative correlation between income inequality and independence—suggesting that more egalitarian societies typically value independence—China presents an anomaly (for more details about the relationship between income inequality and parenting practices, see the next section). Despite significant income inequality (Gini = 0.42), a substantial majority, around 77%, of Chinese parents highly prioritize independence, reflecting a cultural inclination towards fostering self-reliance and personal decision-making among children. This places China in line with more egalitarian societies like Finland and Germany, where similar values prevail despite disparities in income distribution (Gini coefficient around 0.3). As autonomy inherently involves the process of achieving independence [41], our study aims to explore the concept of independence from a broader perspective, specifically focusing on the paradigm of autonomy-granting within the realm of contemporary Chinese parenting practices. Thus, the following hypothesis is made:

**H3.** *Autonomy-granting will be a significant characteristic of parenting during the COVID-19 pandemic*.

### 2.3. Impact of Income Inequality on Parenting

Many qualitative and quantitative studies have found that individual-level factors, such as parental resources, child gender, age, ethnicity, and migration status, can impact parenting practices [7,8,44,45]. However, our study emphasized a provincial-level influence on parenting that has not been thoroughly explored within the Chinese context [44].

Doepke and Zilibotti [10] contended that economic factors stand out as remarkably predictive in determining parental behavior across countries, except for the influence of a country’s political history, ethnic diversity, and cultural context. The reason is that in unequal societies, ensuring favorable outcomes for children is crucial. Academic performance and participation in extracurricular activities play pivotal roles in determining admission to prestigious educational institutions. Lack of access to such opportunities can result in long-term economic hardships and adversity. This assertion opens a window into examining variances in parenting styles within the spectrum of income inequality among different countries. For instance, in Switzerland, known for its low-income disparity (with a Gini coefficient of 0.29 in 2018), there is a tendency towards a more permissive parenting style, emphasizing values such as independence and creativity. This contrasts with the more demanding parenting seen in Turkey (with a Gini coefficient of 0.29 in 2018), where there is a cultural emphasis on diligence. Moreover, within countries, economic trends can influence shifts in parenting styles. As seen in the United States, where the Gini coefficient has increased from 0.43 to 0.49 over the past three decades, there has been a concurrent trend towards more engaged and child-focused parenting practices, reflected in increased parental childcare time, particularly in active childcare activities, and a greater emphasis on structured activities for children [46].

These international and intranational findings regarding the influence of income inequality on parenting provide a compelling framework for our investigation into China’s provincial disparities. As the largest developing country, China has witnessed unparalleled economic growth and a significant poverty reduction [11]. However, there are large economic developmental disparities (Chen et al., 2021) [47]. For instance, in 2021, the per capita net income for Gansu Province was CNY 22,066, while it was CNY 78,027 for Shanghai City [48]. Within provinces, Qinghai Province had the lowest Gini coefficient value of 0.260, while Ningxia region exhibited the highest value of 0.482 (see Appendix A). Such disparities frame our inquiry: How do provincial income inequalities within China influence parenting styles?

Additionally, considering the cultural preferences for boys over girls, the greater childcare responsibilities traditionally assigned to mothers compared to fathers, and the stark urban–rural divide within China, it is plausible that provincial income inequalities might have disproportionately varied effects on these distinct subgroups.

Given the backdrop provided by Doepke and Zilibotti’s [10] argument and specifical unbalanced provincial development and cultural norms in China, we hypothesized that:

**H4.** *In provinces with higher income inequality, there is a tendency towards more demanding and harsh parenting styles compared to provinces with lower income inequality*.

**H5.** *The effects of provincial income inequality differ across various subgroups (i.e., child gender, maternal education level, and hukou status)*.

## 3. Method

### 3.1. Sample

We analyzed data from the 2020 wave of the China Family Panel Studies (CFPS), a biennial survey managed by the Institute of Social Science Survey at Peking University. This fifth wave was conducted from July to December 2020, during which approximately 62,500 interviews were successfully completed across 31 provinces and regions. The survey offers insights into Chinese family life, including community, family, adult, and child data. A team of 554 interviewers across the country was engaged for 176 days. Due to COVID-19, there was a significant shift in data collection methods, with the majority (about 89%) of interviews transitioning from face-to-face to telephone-based interviews [49]. The sample includes a wide range of regions representing the geographic and economic diversity of China. These regions encompass economically developed urban areas such as Beijing, Shanghai, and Guangdong, as well as less developed rural areas in provinces like Gansu and Guizhou. This diverse sampling allows for a comprehensive analysis of how regional differences, such as varying levels of economic development, impact parenting styles.

Our initial sample comprised 4051 children aged 7–16 years in 2020 [50]. However, due to missing values in the parenting-style items, we followed Enders’ [51] recommendation and removed observations with more than 20% missing data. Little’s missing completely at random test confirmed random missing patterns (χ^2^ = 303.01, *df* = 259, *p* = 0.061). This process resulted in a refined sample of 3768 children. Descriptive statistics can be found in Appendix A.

### 3.2. Measures

Parenting: To identify the different parenting styles, we included three theoretical parenting facets: demandingness, responsiveness, and autonomy-granting [52]. We measured demandingness using three questions for parents, e.g., “Do you check your child’s homework?” We measured responsiveness using three questions for parents, e.g., “Do you give up watching TV for your child’s studies?” Finally, we measured autonomy-granting using four questions for children, e.g., “When you make a mistake, your parents seek to understand the reasons, discuss the correct actions with you, and encourage you to learn from the experience”. All items were scored on a five-point scale (1 = very often, 2 = often, 3 = sometimes, 4 = rarely, and 5 = never). Higher scores indicated higher levels of the parenting facets. Cronbach’s alphas for the three subscales were 0.79, 0.84, and 0.81, respectively. Item wording can be found in Appendix A.

Provincial Income Inequality: The Gini coefficient is calculated at the provincial level to measure regional income inequality. In our sample, the Gini coefficients of the included provinces and regions ranged from 0.26 in Qinghai to 0.48 in Guizhou and Ningxia. This variation (mean = 0.42; SD = 0.05) in income inequality across regions allows for an in-depth analysis of how economic disparities influence parenting styles (see Appendix A). However, if there is a large error in the income of middle- and high-income groups, the Gini coefficient estimation may be inaccurate [53]. Therefore, we calculated the Theil index to more comprehensively reflect the income inequality [54] and performed robustness checks, detailed in the next section. Both the Gini coefficient and the Theil index were standardized for running the regression model.

Parental Resources: The CFPS categorizes parental social class into three groups based on occupation: wage job, self-employed, and agricultural work. Parental education was measured by maternal and paternal years of education. We coded parental political resources as 1 if either parent was a member of the Chinese Communist Party (CCP) and 0 otherwise. The CCP, with over 100 million members, represents approximately 7.1% of the population [9]. Party officials of ten occupy higher positions, which are directly linked with power and the control and command of superior resources in China’s socioeconomic structure [9].

Controls: Based on Belsky’s ecological model of parenting [55,56] and a systematic review [24], we selected the following covariates that were potentially related to parenting: sex, hukou status, ethnicity, age, and family size. We coded child gender as male = 1, female = 0; hukou status, a household registration system in China that classifies residents as rural or urban, was coded as rural = 1 and urban = 0. Ethnicity was coded as minority ethnic group = 1 and Han = 0, with Han being the majority ethnic group in China, comprising about 92% of the population. Child age was quantified in years and set family size as the number of individuals residing in a household, intending to partially control for the impact of siblings and grandparents.

### 3.3. Analysis

First, to identify distinct parenting patterns, we employed latent class analysis (LCA), a technique that identifies subgroups within a population based on individuals’ response patterns to observed variables. Instead of using aggregate scores, LCA utilizes the raw item-level data from the demandingness, responsiveness, and autonomy-granting questions. Each item serves as an indicator, and the model classifies individuals into latent classes based on their unique response patterns across these indicators. This approach allows for a data-driven identification of parenting styles, revealing hidden subgroups characterized by similar parenting behaviors.

LCA has gained significant popularity in the educational field for its ability to uncover hidden clusters of people [57]. This technique is adept at identifying subgroups within a larger, varied population, each characterized by similar attributes and exhibiting the same response tendencies [58]. LCA shares the goal of traditional clustering techniques like hierarchical clustering, which seeks to identify groups with consistent reaction patterns to various items [59]. However, what sets LCA apart is its data-driven methodology that assigns individuals to classes based on empirical evidence, potentially leading to more solid and dependable categorization results [60].

We performed LCAs to determine the optimal number of classes that best described the identified parenting styles. The model selection process relied on a set of statistical criteria [61,62], including the Bayesian information criterion (BIC) to assess the model fit, the Lo–Mendell–Robin likelihood ratio test (LMR–LRT) to compare model improvements, entropy to measure the classification precision [63], the principle of parsimony in favor of simpler models, and the imperative of theoretical interpretation to ensure meaningful class identification and practical implications [62].

Next, we examined the relationships between parenting styles, provincial income inequality, and unequal access to higher education using the following model [64]:$$log\left(\frac{\left(p{\left(Y\right)}_{ij}=k\right)}{\left(p{\left(Y\right)}_{ij}=ref\right)}\right)=\beta_{0}+\beta_{1}{Gini}_{j}+\beta_{2}X_{ij}+\gamma_{j}+\varepsilon_{ij}$$
where $Y_{ij}$ represents the parenting style of an individual (i) in a region/province (j), ${p(Y}_{ij}=k)$ denotes the probability that individual i in province j exhibits parenting style k, ${p(Y}_{ij}=ref)$ denotes the probability that individual i in province j exhibits the reference parenting style, ${Gini}_{j}$ denotes the provincial Gini coefficient, $X_{ij}$ denotes individual-level controls, $\gamma_{j}$ is a dummy variable accounting for the systematic differences in parenting styles across provinces, and $\varepsilon_{ij}$ is the error term.

Finally, we explored whether the effects of income inequality on parenting styles have varied across the different subgroups (i.e., child gender, maternal education level, and hukou status). As mothers’ educational attainment notably impacts their children more than that of fathers [65,66], we focused our subgroup analyses on maternal education rather than paternal education. We performed data cleaning and LCAs using R 4.0.3 [67] and assessed the impacts of provincial income inequality using SPSS 27.

## 4. Results

### 4.1. Latent Classes of Parenting

We determined the optimal number of latent classes for parenting through LCAs, testing models with two to five classes (see Table 1). Progressing from two to five classes improved model fit, with higher log-likelihood values and lower BIC values. Entropy values remained consistently high across all models, indicating good class separation. Notably, the three-class model had the highest entropy value, suggesting the most distinct classification at this level. Next, the LMR–LRT test revealed significant differences between the two-class and baseline models (*p* = 0.001). The three-class model showed improvement over the two-class model (*p* = 0.034), but adding more classes (four and five classes) did not significantly enhance the model fit. Therefore, we determined that the three-class model was the most appropriate, providing a balance between fit and parsimony.

When individuals were classified into the three-class model, the parenting typologies could be defined according to the patterns of the conditional item-response probabilities. Figure 1 shows the estimated sizes and conditional probabilities of the identified classes. The three latent classes accounted for 48.2%, 27.7%, and 24.1% of the sample, respectively. Regarding the conditional item-response probabilities, class 1 parents were very strict and responsive, and allowed a high level of autonomy-granting to their children, suggesting an authoritative parenting style. Conversely, relative to the class 1 parents, those in class 2 placed more emphasis on autonomy-granting while demonstrating lower levels of demandingness and responsiveness, suggesting an autonomy-granting parenting style. The latent class 3 scores were consistently moderate across all items, indicating an average-level undifferentiated parenting style. This style does not exhibit notably high or low tendencies in any specific facet, distinguishing it from neglectful parenting, which generally scores particularly low in each facet.

### 4.2. Inequality in Parenting Styles

We further assessed the influence of the Gini coefficient on the identified parenting styles (see Table 2). The odds ratio (OR) of 1.704 indicates that, for each standard deviation (SD) increase in the Gini coefficient, the probability of parents choosing autonomy-granting over average-level undifferentiated parenting increased by approximately 70.4%. Furthermore, this increase in the Gini coefficient doubled the likelihood of parents adopting authoritarian parenting over average-level undifferentiated parenting (OR = 2.193 ***). However, the Gini coefficient did not significantly affect the choice between autonomy-granting and authoritarian parenting (OR = 0.968).

Other covariates are discussed in the Appendix A.

### 4.3. Subgroup Analyses

The effects of the Gini coefficient on parenting styles may vary across subgroups. Thus, we conducted further analyses by dividing the sample into the following subgroups: child gender, maternal education level, and hukou (rural vs. urban). The results are presented in Table 3.

For male children, one SD increase in the Gini coefficient reduced the likelihood of choosing autonomy-granting over authoritarian parenting by approximately 22% (OR = 0.783 **). Conversely, it increased the likelihood of adopting autonomy-granting over average-level undifferentiated parenting by 93% (OR = 1.931 ***). Moreover, males were about 2.83 times more likely to experience authoritarian versus average-level undifferentiated parenting in relation to one SD increase in the Gini coefficient (OR = 2.832 ***). In contrast, for female children, one SD increase in the Gini coefficient merely increased the chances of authoritarian over average-level undifferentiated parenting by about 41% (OR = 1.411 **).

Regarding maternal education, mothers with less than 12 years of education were approximately 2.6 times more likely to choose authoritarian over average-level undifferentiated parenting (OR = 2.601 ***). In rural regions, an SD increase in the Gini coefficient significantly increased the likelihood of opting for authoritarian parenting over average-level undifferentiated parenting by about 2.189 times (OR = 2.189 ***). In urban settings, a higher Gini coefficient was associated with a 1.395 times greater probability of selecting authoritarian over average-level undifferentiated parenting (OR = 1.395 ***). Non-significant values were not discussed.

### 4.4. Robustness Checks

To ensure the robustness of our findings, we conducted two sensitivity analyses. First, we re-executed the latent class analysis excluding non-parent respondents who completed the questionnaire, which accounted for 6.2% of the responses. The results from this modified analysis aligned with those from the complete dataset, affirming the consistency of our findings (see Appendix A).

Second, we replaced the Gini coefficient with the Theil index to measure income inequality (see Appendix A). Only one alternative significantly altered the point estimates related to income inequality.

## 5. Discussion

Analyzing data from 3768 children across 25 Chinese provinces during the first year of the COVID-19 pandemic, we identified three predominant parenting styles: authoritarian (48.2%; H1 supported), autonomy-granting (27.7%; H3 supported), and average-level undifferentiated (24.1%; H2 rejected). We then probed the influence of provincial income inequality on these parenting styles, uncovering that the increase in the Gini coefficient significantly impacted the prevalence of autonomy-granting and authoritarian styles (H4 partially supported). Furthermore, subgroup analyses revealed that the effects of provincial income inequality vary depending on child gender, maternal education level, and hukou status, illustrating the nuanced interplay between economic conditions and parenting during an unprecedented global health crisis (H5 supported).

### 5.1. Parenting Styles in China

Authoritarian Parenting: Not surprisingly, authoritarian parenting emerged [68]. However, a distinctive aspect of our study lies in the notable discrepancy in the prevalence rate of authoritarian parenting, which stood at 48.7%, surpassing the typically observed rates of less than 25% in prior research [9,21,69]. The reasons could be, first, related to measurement items. Authoritarian parenting in our study exhibited moderate levels of responsiveness, deviating from the conventional understanding outlined by Baumrind [19]. The inclusion of items assessing responsiveness, such as “monitoring homework completion” and “sacrificing leisure time for the child’s academic pursuits,” introduced overlap with measures of demandingness, potentially impacting our findings. Moreover, the unprecedented circumstances precipitated by the COVID-19 pandemic have brought about multiple changes in daily life, compounded by China’s stringent containment measures. These factors have contributed to heightened stress and uncertainty among parents, consequently amplifying their expectations and demands placed upon their children.

Autonomy-Granting Parenting: Our study innovatively integrated the autonomy-granting dimension into the measurement of parenting styles. While this dimension has been discussed theoretically [40], our study represents one of the first attempts to empirically examine it. Autonomy-granting parenting in our study not only showed a high level of autonomy support but also included significant levels of demandingness and responsiveness. The emergence of autonomy-granting parenting can be attributed to socio-economic transformations and evolving cultural values in contemporary China. Additionally, self-determination theory (SDT [70]) posits that fulfilling the need for autonomy is crucial for enhancing effective self-regulation and well-being. This theoretical perspective supports the relevance of incorporating autonomy-granting as a dimension in our assessment of parenting styles, underscoring its importance in fostering an environment conducive to developing autonomy in children even in a challenging environment.

This finding holds a significant practical implication. Prior studies have theoretically argued that the dimensions of demandingness and responsiveness may not fully encapsulate the breadth and depth of parenting dynamics, such as warmth and inductive reasoning [21,71]. However, selecting dimensions that aptly and concisely reflect the research context is frequently overlooked. Our research serves as a compelling illustration, underscoring the necessity of incorporating dimensions that capture the characteristics of the era—such as autonomy-granting—to gain a deeper understanding of parental behaviors within the evolving socio-economic and cultural landscapes of China.

Average-Level Undifferentiated Parenting: The finding of average-level undifferentiated parenting, which displayed no particularly distinct characteristics, was an unexpected finding that did not align with our initial hypotheses. This parenting style is characterized by a balanced and moderate approach, differing from neglectful parenting in its moderate levels of engagement across parenting domains. From a cultural perspective, this average-level undifferentiated parenting can be seen as a reflection of traditional Chinese values that prioritize balance and harmony. In Chinese culture, moderation and avoiding extremes are often considered virtues, influenced by Confucian and Taoist philosophies [72]. These cultural values may lead parents to adopt a more balanced approach to parenting, aiming to provide a stable and supportive environment without exerting excessive pressure or neglecting their children’s needs. This style is akin to the “average” parenting concept identified by Nelson et al. [71] and bears similarity to Baumrind’s [19] notion of good-enough parenting.

Absence of Neglectful, Permissive, and Authoritative Parenting: Our findings reveal a complete absence of neglectful and permissive parenting style, inconsistent with prior studies [9,69] and exceeding the hypothesized reduction. One plausible reason is the strong emphasis on the parental role as an educator within Chinese culture. In Chinese culture, indulgence and neglect towards children are seen as an abdication of parental responsibilities, which is not encouraged by traditional values [21]. Additionally, Chinese parents have high expectations for their children’s academic achievements. To meet these expectations, Chinese parents are deeply involved in their children’s academic activities, resulting in permissive and neglectful parenting being rare or even nonexistent [43]. The final reason would be the intense familial interactions enforced by the extensive and strict lockdown measures during the pandemic in China. As previously discussed, the Chinese government implemented more stringent quarantine measures than other nations. These conditions likely compelled even typically less engaged parents to become more actively involved in their children’s daily activities [73], effectively eliminating neglectful and permissive parenting behaviors. This total absence underscores the significant impact of the pandemic on altering the common parenting behaviors in China.

Furthermore, our study did not identify the authoritative parenting style typically prevalent in the literature. A plausible reason could be that the identified autonomy-granting parenting in our study not only exhibits a high level of autonomy-granting but also demonstrates significant levels of demandingness and responsiveness, which are characteristic traits of authoritative parenting. As a result, some facets of authoritative parenting may have been inadvertently reflected in autonomy-granting parenting.

### 5.2. Provincial Income Inequality and Parenting

Our study concurrently substantiates Doepke and Zilibotti’s [10] observation that a significant inclination towards authoritarian parenting emerged in provinces with high income inequality, which reflects the desire to provide a structured and nurturing environment that provides children with the essential skills needed to navigate socioeconomic challenges. However, our results challenge the proposition that areas with low income inequality tend to favor a permissive parenting style. This divergence can be attributed to China’s cultural emphasis on education [74], where, even in less economically disparate regions, the value placed on academic success persists. This Chinese cultural norm underscores the importance of education as the key to social mobility and success [75], influencing parenting practices across various socioeconomic backgrounds.

Except for a preference for authoritarian parenting, we have a new finding: Higher income inequality was also associated with a greater propensity toward choosing autonomy-granting than average-level undifferentiated parenting. Self-determination theory posits that autonomy is a fundamental psychological need [70]. When children’s autonomy needs are well supported, they exhibit greater autonomy in their behaviors and are more likely to persist in their pursuits. Thus, when facing income inequality, parents adopt an autonomous parenting style to empower their children to develop these essential skills, thereby enabling them to navigate the challenges and opportunities presented in a dynamic socioeconomic landscape.

Interestingly, we did not find a significant difference between autonomy-granting and authoritarian parenting considering the income inequality. This suggests that parents might oscillate between authoritarian and autonomy-granting styles. In regions with high income inequality, authoritarian parents exert control to ensure that children adhere to behaviors deemed beneficial for coping with adversity. Meanwhile, autonomy-granting styles promote independence and autonomy, fostering self-reliance and decision-making skills crucial for navigating unpredictable conditions [70]. Both reflected that parents adapt their parenting style based on their assessment of environmental challenges and their child’s needs [76].

### 5.3. Divergent Parenting in China

We found varying effects of the Gini coefficient across the different subgroups, as follows:

Child Gender: In regions with greater income inequality and fewer spots available at top universities, parents are more likely to prefer authoritarian parenting first, then autonomy-granting, and finally average-level undifferentiated parenting for their sons. For female children, the results indicated that high provincial income inequality modestly nudges parents towards authoritarian parenting.

These divergent gender influences aligned with the traditional Chinese cultural norms that prioritize sons as the primary recipients of educational and developmental resources [77]. Historically, Confucian values have deeply influenced Chinese society, emphasizing the importance of males as the primary earners and bearers of the family lineage. In this context, sons are often viewed as responsible for supporting their parents in old age and achieving high social and career status. This cultural expectation is reinforced by longstanding societal structures and government policies that have favored male education and employment opportunities. Under these traditional gender norms, males are considered the primary earners for the family, and there are higher expectations for their career and social status success [78]. Consequently, as income inequality widens, families tend to harbor greater expectations for their sons and are more likely to engage in authoritarian parenting practices, aiming to enhance their sons’ prospective success. This includes strict discipline, high academic expectations, and limited autonomy to ensure that sons are well prepared to compete in a highly unequal economic landscape.

Maternal Education Level: Our findings revealed that mothers with a high-school education or lower displayed a stronger preference for authoritarian parenting over average-level undifferentiated parenting as the Gini coefficient increased. This suggests that in socioeconomically strained environments, less educated mothers tend to adopt a demanding parenting style to support their children’s development and education [79]. In contrast, mothers with higher educational levels showed a slight decrease in their inclination towards authoritarian parenting. This may be attributed to their greater access to knowledge and resources, allowing them to utilize diverse resources less influenced by economic stressors [80].

Hukou (Rural vs. Urban): Education in China shows a clear divide between urban and rural areas. Historically, the hukou system has played a significant role in this disparity. Established in the 1950s, the hukou system categorized residents as either urban or rural, with profound implications for their access to public services, including education, healthcare, and employment opportunities. Rural hukou holders were often restricted to their registered areas, limiting their mobility and access to resources compared to their urban counterparts [81]. Currently, rural residents still have fewer resources and opportunities due to a combination of economic factors and government policies. For instance, government investment in education has historically favored urban areas, resulting in better school facilities, more qualified teachers, and greater educational resources in cities. In contrast, rural schools often suffer from inadequate funding, poor infrastructure, and a shortage of qualified teachers. This urban–rural divide has led to significant gaps in educational quality and access, contributing to lower educational outcomes for rural children.

In agreement with these insights, we found that in rural areas, the high income inequality significantly impacted the choice of authoritarian over average-level undifferentiated parenting. These findings suggest that in response to the growing awareness of the critical role of parenting in the development of human capital, more rural families are adopting more structured, authoritarian parenting approaches, which can help to bridge the gap between urban–rural segregation and potentially reduce the educational disparities between these areas [82]. Conversely, in the urban residency subgroup, income inequality showed a less significant effect on parenting styles. This may indicate that a more resource-abundant urban environment can shield families from the pressures of economic variability and educational access to a greater extent than those experienced in rural areas [83]. Therefore, compared to their rural counterparts, urban parents may not experience the same need to adjust their parenting styles in response to economic and educational changes.

## 6. Limitations

Our study has the following limitations. First, with a sample size of 3768 children, the generalizability of the results to the Chinese population remains uncertain. Future research should use a larger, more representative sample that encapsulates China’s vast demographic diversity. Second, the use of a cross-sectional dataset restricted our ability to observe changes in parenting styles over time. Post-COVID-19, parenting approaches may be evolving (e.g., neglectful parenting may potentially disappear), so longitudinal studies are crucial for capturing these dynamic shifts. Last, we only measured three facets of parenting, which may not fully capture the complexity of parenting styles. Prior research has highlighted the importance of additional dimensions, such as harshness and the encouragement of achievement in parenting (e.g., Zhang et al. [20]). Future studies should include a broader range of parenting facets to provide a more comprehensive understanding of parenting practices.

## 7. Implications

Despite these limitations, this study has several significant implications. Theoretically, our study expands the discourse by incorporating a previously neglected aspect of parenting: autonomy-granting. Although it has been discussed theoretically, this study is among the first to empirically test its impact. The findings reveal that autonomy-granting is a crucial characteristic of parenting even during crises. Parental roles are deeply intertwined with cultural expectations and societal norms, which emphasize academic success and filial piety [21,43]. This cultural context influences parenting styles significantly, as seen in the prevalence of authoritarian parenting in the provinces with high income inequality. However, as societies modernize and educational opportunities expand, there is a gradual shift towards recognizing the importance of nurturing children’s individual abilities and promoting independent thinking [41,43]. Thus, the emergence of autonomy-granting parenting may indicate the evolving socio-cultural values in contemporary China, where the authoritarian approaches are being complemented by more autonomy-supportive practices.

From a practical standpoint, our findings have both national and international implications. Given the crucial role of parenting styles in the development of adolescents’ human capital [84], the relevant authorities and media platforms should encourage greater parental involvement in youth education. For parents who lack knowledge of child-rearing, providing accessible educational resources, especially for disadvantaged families constrained by financial and time limitations, can improve human capital levels in such households and mitigate educational inequalities stemming from disparities in educational investment [85]. Also, more interventions are needed to address the structural inequalities embedded in the hukou system and increase investment in rural education to reduce disparities and support more balanced development across regions.

Internationally, for Confucian countries that share similar cultures with China (e.g., Japan and South Korea), the interplay between socioeconomic factors and parenting styles may exhibit comparable dynamics [86,87]. In these countries, where men often hold dominant societal positions, a strong market orientation prevails, and educational attainment is highly valued [88,89], similar patterns of authoritarian and autonomy-granting parenting styles may emerge based on socioeconomic status and educational background. For instance, in high-income inequality areas within these countries, parents might exhibit authoritarian parenting to ensure their children’s success in a highly competitive academic environment.

The United Nations Children’s Fund (UNICEF, 2020) has recommended several core standards for parenting programs, including tailoring content to the child’s developmental stage, serving vulnerable children and their families, involving all parents and key caregivers, adapting to cultures and contexts, and building positive parenting practices [90]. However, imposing an intervention developed elsewhere on a new population may do more harm than good; moreover, some unintended consequences can be anticipated and prevented, whereas others cannot [59]. Policymakers could carefully consider implementing regional-specific educational and social programs to address the unique challenges faced by families in various socioeconomic contexts. Therefore, targeted interventions, such as providing financial support, educational resources, and parenting programs in high income inequality regions, and offering parenting education and support services in rural areas, are essential to promote positive parenting practices. Gender-sensitive interventions and support for mothers with lower educational levels are also critical to addressing the disparities in parenting styles. By adopting these tailored approaches, policymakers can create more equitable and supportive environments for families, ultimately contributing to better outcomes for children across the country.

## Figures and Tables

**Figure 1 behavsci-14-00587-f001:**
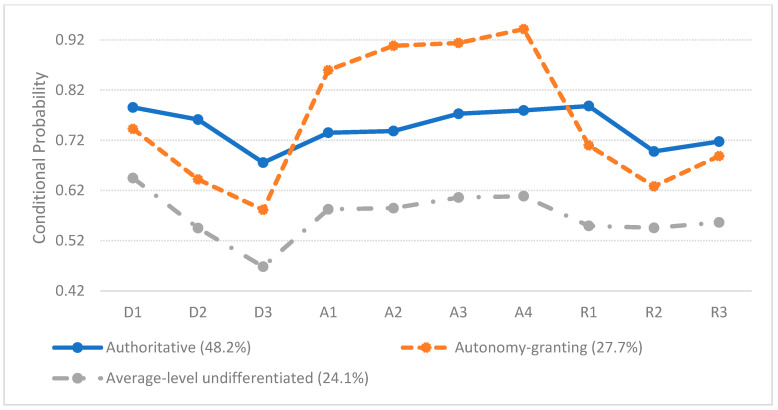
Latent classes of parenting. Note: Indicators measured three distinct facets of parenting: demandingness (D1–D3), autonomy-granting (A1–A4), and responsiveness (R1–R3).

**Table 1 behavsci-14-00587-t001:** Fit statistics for LCAs modeling the parenting styles.

Class	Loglik	df	BIC	Entropy	p (LMR–LRT)
2	−61,361.7	6669	123,808.6	0.85	0.001
3	−59,082.1	6607	119,796.6	0.9	0.034
4	−57,349.7	6545	117,878.9	0.79	0.331
5	−56,251.5	6483	116,229.4	0.77	0.218

Note. BIC, Bayesian information criterion; ABIC, adjusted Bayesian information criterion; LMR–LRT, Lo–Mendell–Rubin likelihood ratio test, and entropy value testing classification quality.

**Table 2 behavsci-14-00587-t002:** Odds ratios for regression analysis of parenting styles (*N* = 3768).

	Autonomy Granting vs. Authoritarian	Autonomy Granting vs. Average-Level Undifferentiated	Authoritarian vs. Average-Level Undifferentiated
Gini coefficient	0.968 (0.384)	1.704 *** (0.201)	2.193 *** (0.213)
Paternal education	1.124 * (0.103)	1.118 * (0.198)	1.139 * (0.151)
Maternal education	1.218 ** (0.115)	1.114 * (0.261)	1.130 * (0.138)
Parental CCP	1.579 ** (0.117)	1.427 * (0.213)	1.189 (0.279)
Parental class (ref: agricultural work)			
Wage job	1.356 ** (0.191)	1.709 *** (0.268)	1.026 (0.278)
Self-employed	1.304 ** (0.074)	1.575 *** (0.178)	1.232 (0.175)
Male	0.982 (0.265)	1.309 *** (0.247)	1.445 *** (0.267)
Child age	1.102 (0.192)	1.280 ** (0.142)	1.417 *** (0.188)
Rural	0.647 *** (0.107)	1.123 (0.201)	1.476 *** (0.138)
Family size	0.519 *** (0.095)	0.677 *** (0.109)	0.965 (0.165)
Minority	0.433 *** (0.089)	0.539 ** (0.124)	1.186 (0.294)

Note: Exponentiated coefficients; standard errors are in parentheses. *** *p* < 0.001, ** *p* < 0.01, * *p* < 0.05.

**Table 3 behavsci-14-00587-t003:** Subgroup analyses of the impact of the Gini coefficient and Project 211 admission quotas on parenting styles.

	Autonomy Granting vs. Authoritarian	Autonomy Granting vs. Average-Level Undifferentiated	Authoritarian vs. Average-Level Undifferentiated
Male (*n* = 1959)	0.818 *** (0.113)	1.931 *** (0.215)	2.832 *** (0.194)
Female (*n* = 1809)	1.079 (0.240)	1.317 (0.289)	1.411 ** (0.132)
Maternal education <12 years (*n* = 2318)	0.987 (0.083)	1.102 (0.133)	2.601 *** (0.079)
Maternal education >12 years (*n* = 1315)	1.022 (0.287)	1.447 *** (0.225)	1.635 *** (0.173)
Rural (*n* = 1997)	1.065 (0.228)	1.095 (0.121)	2.189 *** (0.178)
Urban (*n* = 1771)	1.065 (0.101)	1.188 (0.298)	1.395 *** (0.198)

Note: Exponentiated coefficients; standard errors are in parentheses. *** *p* < 0.001, ** *p* < 0.01.

## Data Availability

Publicly available datasets were analyzed in this study. This data can be found at: https://www.isss.pku.edu.cn/cfps/en/index.htm (accessed on 8 July 2024).

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
