# Peer review of "Impact of Provincial Income Inequality on Parenting Styles in China during COVID-19"

_behavsci, 2024, doi:10.3390/bs14070587_

Round 1

Reviewer 1 Report

Comments and Suggestions for Authors

Dear Authors!
This valuable endeavour is relevant not only for research, but also has practical implications. The cross-sectional study design is appropriate and the sample size is adequate.

However, there are some points that need more reflection, completion or clarification.

1. There is quite much unclarity with respect to the parenting styles, The three parenting styles introduced in the literature review were authoritative, authoritarian, and permissive (the fourth being the neglectful one). The autonomy-granting type is introduced later on a theoretical level,  and also found in the empirical analysis. In the Discussion, the lack of the authoritative style and neglectful types is coming as a surprise, but there is no argument for/against the permissive style. This suggests that the autonomy-granting style is similar or the same as the permissive style. Is this the case? Please argue. In short, there are six parenting styles, some of them ommitted, others articulated, which confuses the reader.

2. The description of Chinese regions are missing, although the title itself promises this aspect, and the differences in the inequalities demand addressing this issue of regions and regional differences in more detail. The Implications for practice also state that different regions have distinct parenting styles (line 535), but this was not discussed at all. 

3. Third, there is no evidence that parenting styles have changed due to the lockdown. The authors state it in several instances that the lockdown have changed parenting styles, but due to the cross-sectional study design this remains an assumption. 

3. Methodologically, there is need for more details on how parenting styles were operationalized. Examples of questions are provided, but how the scores are computed for the different aspects is missing. 

4. It is quite important to note that there are some slangs which are unfamiliar to readers outside of China, like CCI member or Han, without any clarification to that. Hukou status is also nor commonly used in other cultures. Please complete.

5. Some of the hypotheses cannot be varified, and do not relate to associations. Only hypotheses 4 and 5 take the classical form of a varifiable assumption.

Major revisions are needed for this paper to be reconsidered. Good luck!

With best wishes,

reviewer

Author Response

Dear Reviewer, thank you very much for your great feedback. Attached is our response.

Reviewer 2 Report

Comments and Suggestions for Authors

Comments on the Quality of English Language

I had some suggestions for you in the file above.

Author Response

Thank you very much for your kind suggestions. We greatly appreciate your feedback and have accepted most of your recommendations.

Reviewer 3 Report

Comments and Suggestions for Authors

Wednesday, 19 June 2024

Given the vast range of socioeconomic and cultural settings across the provinces of Mainland China, this study arguably fills an apparent gap in the scholarly literature. Our concerns with the paper largely rest on the question of the purpose of a study such as this one. Research publications analogous to this proposed study largely exist as very carefully discerned and carefully well-constructed collections of data, and while we understand the way in which the facts-as-valid-studies school of thought can pervade the relevant scholarly literature of the behavioral sciences, we still openly wonder if the authors might elaborate on the "implications for practice" section from lines 518 through 546---particularly through a discussion on the international implications of this study. Although the authors appear content to confine conclusions to implications in Mainland China, the authors would certainly have a larger audience if the paper broadened the geographical horizons of the study. The authors appear to implicitly acknowledge the validity of this last point by mentioning recommendations of the United Nations Children's Fund (lines 537~539).

Of course, in order to broaden the possible audience of scholars who might express a profound fascination with the topic, the authors of this study will have to explore ways of introducing the Mainland Chinese context to individuals who will, in all likelihood, lack the prerequisite knowledge of Mainland Chinese culture that the authors appear to assume. In the "Divergent Parenting in China" section (4.3), for instance, the authors speak of "traditional gender norms" (line 475), but we would need to know what the authors precisely mean by such a term, since the word "traditional" can have a veritable constellation of definitions across time and place. We would want to know the precise reasons for why, for instance, "traditional gender norms" in Mainland China demand that the role of "primary earners" (line 475) rests with male individuals; some engagement with scholarly literature and/or government policy decisions (if applicable and necessary) would help to acquaint the audience with the nuances of the discussion. We also wonder about why the authors assume that we (as the readers) simply accept the judgment of rural residents having "fewer resources and opportunities" (line 490) at face value---perhaps the authors of the study would do well to explain why this situation has arisen (again, with relevant scholarly literature and/or policy statements).

We also wonder about certain statements that, at least to audiences untutored in the Mainland Chinese context, can come across as sweeping generalizations that do not withstand scrutiny---or, at the very least, require elaboration. The statement of how "these substantial shifts (i.e., the strictness of Mainland Chinese authorities in lines 127~131) have profoundly influenced the practices and psychological approaches of Chinese parents" (lines 132~133) certainly requires a greater degree of elaboration on behalf of the authors; we feel that in this case, the authors seem to have simply assumed the reader's acquiescence to these fascinating and provocative (but ultimately slightly unsubstantiated) claims. Other such instances of this writing arise in [1] lines 113~116 (we wonder about how, for instance, "the increased presence of parents allowed for deeper engagement with children"---the authors give a rather pithy treatment of this topic), [2] lines 99~100 (when the authors simply want us to accept the notion of the dominance of authoritative/authoritarian parenting styles in Mainland China---the authors direct us to a literature review in endnote number 22, but the authors would probably do better to integrate this material into the main body text of the manuscript), and lines 88~96 (we could have benefited from any sort of discussion of Baumrind-parenting-style studies in Mainland China as a way of acquainting us with the ways in which other scholars have sought to see Baumrind-parenting styles among the inhabitants of Mainland China)----and other areas of the draft manuscript on top of these three references of lines 88~96, 99~100, and 113~116. The authors would do well to carefully reread the draft manuscript and then choose (at the free discretion of the authors) about half a dozen more general statements that might require more elaboration through engagement with the scholarly literature.

Author Response

Dear Reviewer, thank you very much for your comments. Attached contains our responses. Due to an operational error, we uploaded two attachments. Only one of them contains our responses (as indicated within the document). We sincerely apologize for any inconvenience this may have caused.

Round 2

Reviewer 1 Report

Comments and Suggestions for Authors

Dear Authors!
The paper has been improved considerably, and is not much more understandable to non-Chinese readers.

Congratulations! In line 570 please replace Child Sex with Child gender.

Best,

reviewer

Comments on the Quality of English Language

In line 570 please replace Child Sex with Child gender.

Author Response

Thank you for your kindly suggestions. We have carefully revised our manuscript and replaced all "child sex" with "child gender".  For example, you can see these changes on lines 77, 85, and 553. -Rui Jin

Reviewer 3 Report

Comments and Suggestions for Authors

The three authors have collectively sought to address the deficiencies indicated in the first round of reviews, and we appreciate the attempts of the authors to introduce some more cultural discussions (the so-called "strict affectionate" notation on line 114 comes to mind in the revisions) in order to accommodate the sensitivities of audiences not necessarily fully aware of the Mainland Chinese context. We also appreciate the expansion of the implications section.

Author Response

Thank you for your valuable suggestions. We have made the following changes to our manuscript:

  1. We have added more cultural explanations of "strict-affectionate" parenting, which emphasizes a balance of strictness and kindness in parenting. Additionally, we compared this concept with tiger parenting (Kim et al., 2013), which may be more familiar to readers. Please see lines 116-125.

  2. In lines 490-496 of the discussion section, we have further elaborated on average-level undifferentiated parenting by providing additional cultural insights.

  3. In lines 625-635 of the implication section, we added more details about autonomy granting that may reflect evolving socio-cultural values in contemporary China. 

Thank you very much.

-Rui Jin